# A Neuro-inspired Interpretation of Unlearning in Large Language Models through Sample-level Unlearning Difficulty

## Abstract

Driven by privacy protection laws and regulations, unlearning in Large Language Models (LLMs) is gaining increasing attention. However, current research often neglects the interpretability of the unlearning process, particularly concerning sample-level unlearning difficulty. Existing studies typically assume a uniform unlearning difficulty across samples. This simplification risks attributing the performance of unlearning algorithms to sample selection rather than the algorithm's design, potentially steering the development of LLM unlearning in the wrong direction. Thus, we investigate the relationship between LLM unlearning and sample characteristics, with a focus on unlearning difficulty. Drawing inspiration from neuroscience, we propose a Memory Removal Difficulty (MRD) metric to quantify sample-level unlearning difficulty. Using MRD, we analyze the characteristics of hard-to-unlearn versus easy-to-unlearn samples. Furthermore, we propose an MRD-based weighted sampling method to optimize existing unlearning algorithms, which prioritizes easily forgettable samples, thereby improving unlearning efficiency and effectiveness. We validate the proposed metric and method using public benchmarks and datasets, with results confirming its effectiveness.

## 1 Introduction

Large Language Models (LLMs) excel at generating human-like text, leading to their broad adoption in various applications. This success largely stems from their strong memorization of the training corpus (Zhang et al., 2023). However, such memorization also raises serious concerns, including risks of privacy breaches (Kim et al., 2024), bias propagation (Yu et al., 2023; Motoki et al., 2024), and the generation of illegal content (Karamolegkou et al., 2023). In particular, privacy laws like the *GDPR* require service providers to remove private information from training data upon user request (Voigt & Von dem Bussche, 2017). This creates a significant challenge: how to effectively erase the influence of specific data samples (i.e., *the forget set*), or higher-level data concepts from pre-trained LLMs.

A practical approach to addressing the issue above is Machine Unlearning (MU) (Liu et al., 2024c). Previous research (Ginart et al., 2019; Ullah et al., 2021; Thudi et al., 2022; Liu et al., 2024a) has primarily focused on MU in classification models, where retraining on the remaining data (i.e., *the retain set*) is the gold standard. However, given the massive scale of training data and the extensive number of parameters in LLMs, this unlearning approach becomes infeasible for LLMs. Therefore, developing effective and efficient methods for implementing MU in LLMs represents a critical challenge that requires resolution.

Existing studies (Jang et al., 2023; Ji et al., 2024a; Feng et al., 2024; Liu et al., 2024c) defines LLM unlearning as the removal of specific knowledge from the forget set (i.e., *unlearning completeness*) while preserving the model's performance on unrelated tasks (i.e., *model utility*). Current methods achieving this can be broadly classified into three categories, i.e., gradient-based methods (Jang et al., 2023; Yao et al., 2024), preference optimization-based methods (Maini et al., 2024; Zhang et al., 2024), and model weight-based methods (Jia et al., 2024). Despite recent advancements, the interpretability of the unlearning process in LLMs remains underexplored. *The lack of interpretability hinders the capability to comprehensively evaluate the practical effectiveness of existing LLM*

*unlearning algorithms.* For instance, the superior performance of certain unlearning algorithms might be attributed merely to the inherent ease of unlearning the selected samples, rather than to any genuine advantage of the algorithms themselves. Such a lack of fine-grained analysis could potentially impact the reliability and generalizability of LLM unlearning algorithms.

Recent studies increasingly explore the interpretability of MU. Fan et al. (Fan et al., 2024) analyze how different partitions of the forget sets influence model performance on the retain sets in image classification tasks. Zhao et al. (Zhao et al., 2024) investigate the presence of explainable features within the forget sets and their impact on the difficulty of unlearning. Chen et al. (Chen et al., 2024) provide a more fine-grained perspective, showing that in recommendation systems, unlearning difficulty varies significantly across users, with potential implications for the evaluation of unlearning algorithms. Collectively, these studies highlight a trend toward sample-level analysis in unlearning interpretability. However, notable limitations remain. These works lack a formal definition of unlearning difficulty at the sample level and offer little theoretical insight into why certain samples are harder to unlearn. Additionally, methods developed for image classification may not effectively generalize to LLMs, which struggle with modeling structured features due to their text-based autoregressive nature. To address these issues, this paper investigates the LLM unlearning problem, focusing on the following three key questions:

- **Q1**. How to design a reasonable and computationally efficient metric to measure the unlearning difficulty of individual data samples?
- **Q2**. Based on this metric, what characteristics make certain samples more difficult to unlearn?
- **Q3**. Can this metric enhance the effectiveness and efficiency of LLM unlearning algorithms?

To address the questions above, this paper undertakes the following contributions:

**To address Q1,** we propose a metric, Memory Removal Difficulty (MRD), to measure the unlearning difficulty of individual samples (e.g., sentences) in LLMs. Inspired by findings in neuroscience (Kim & Fanselow, 1992; Squire & Alvarez, 1995; Frankland & Bontempi, 2005; Konrad et al., 2011), where long-term memories in human brain are typically resistant to minor brain injuries and are not easily forgotten, MRD models unlearning difficulty in LLMs. As shown in Figure 1, it is defined as the expected change in the logit of a sample before and after parameter perturbations, ensuring both reasonable and computational feasibility.

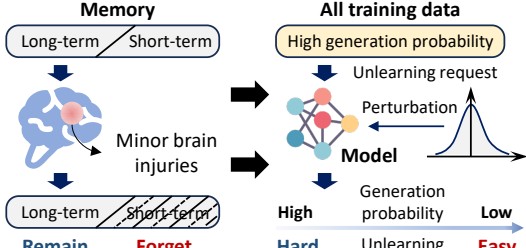

Figure 1: Unlearning difficulty is measured by introducing small perturbations to model parameters (akin to minor brain injuries) and comparing the change in generation probability for a specific sample before and after perturbation.

**To address Q2,** we conduct an in-depth analysis on MRD metric to uncover the characteristics of data samples that make them more difficult to unlearn. For instance, we find that samples with high frequency or those with strong contextual associations to others are often harder to unlearn. Through theoretical analysis and experimental validations, we provide clear explanations for these properties, thereby offering insights into the factors influencing the unlearning difficulty.

**To address Q3,** we propose an MRD-based weighted sampling method to optimize existing unlearning algorithms. Inspired by curriculum learning, MRD serves as a scoring function to adjust the sampling probability of unlearning samples, enabling a dynamic progression from simple to complex unlearning sequences. Comparative experiments demonstrate that this method significantly accelerates convergence and improves performance, highlighting MRD as an effective measure of unlearning difficulty and a practical tool for optimizing unlearning algorithms.

## 2 RELATED WORK

### 2.1 MACHINE UNLEARNING

MU methods can be categorized into exact unlearning and approximate unlearning (Xu et al., 2023). Exact unlearning methods treat the retrained model as the gold standard to achieve complete erasure of the target data. These methods divide the model or dataset into multiple sub-components and

construct an ensemble system, thereby distributing the computational overhead of retraining across these sub-components during the unlearning process (Bourtoule et al., 2021; Li et al., 2024b). In contrast, approximate unlearning methods aim to obtain a model that is approximately equivalent to the retrained model in terms of either model parameters or outputs. These methods are typically achieved by estimating the influence of the target data (Koh & Liang, 2017; Liu et al., 2024b) or by fine-tuning a defined objective function.

## 2.2 LLM Unlearning

LLM unlearning is typically framed as approximate unlearning, aiming to achieve both high unlearning completeness and model utility. Jang et al.(Jang et al., 2023) first propose a gradient ascent method on the forget set, which significantly improves unlearning completeness but at the cost of reduced model utility. To mitigate this, subsequent studies (Maini et al., 2024; Yao et al., 2024) introduce regularization-based enhancements (e.g., parameter and loss regularization). However, these methods still face challenges in balancing the trade-off between completeness and utility. Later studies (Zhang et al., 2024) approach unlearning by treating the forgotten data as negative examples in preference alignment, formalizing the process as a preference optimization task with predefined positive responses (e.g., refusals or counterfactual samples). While this integrated optimization approach has shown some success, it suffers from low unlearning efficiency, limiting its practicality. Recent research (Jia et al., 2024) revisits the problem through model weights, leveraging the modular structure of LLMs to identify and guide unlearning at the module level. Although this method provides valuable insights, its computational efficiency remains low, posing significant challenges for real-world applications.

## 3 Interpretability of LLM Unlearning

### 3.1 Problem Setup of LLM Unlearning

**Autoregressive Model Training.** Given a training set $\mathcal{D} = \mathcal{D}_F \cup \mathcal{D}_R$, where $\mathcal{D}_F = \{\boldsymbol{x}^1, \boldsymbol{x}^2, \ldots, \boldsymbol{x}^{N_f}\}$ and $\mathcal{D}_R = \{\boldsymbol{x}^1, \boldsymbol{x}^2, \ldots, \boldsymbol{x}^{N_r}\}$ represent the forget and retain sets with $N_f$ and $N_r$ samples, respectively, each sample $\boldsymbol{x}^i = \{x_1, \ldots, x_{n_i}\}$ corresponds to a sample of length $n_i$. The parameters $\boldsymbol{\theta}'$ of a model autoregressively trained on $\mathcal{D}$ satisfy the following equation:

$$\boldsymbol{\theta}' = \arg\min_{\boldsymbol{\theta}} \mathcal{L}_{NLL}(\mathcal{D}; \boldsymbol{\theta}) = \arg\min_{\boldsymbol{\theta}} -\mathbb{E}_{\boldsymbol{x}^i \sim \mathcal{D}} \left[ \sum_{t=1}^{n_i} \log p(x_t \mid \boldsymbol{x}_{<t}; \boldsymbol{\theta}) \right]. \quad (1)$$

**Objective of LLM Unlearning.** To unlearn a sample $\boldsymbol{x}^i$, the objective is typically formalized as the following optimization problem (Jang et al., 2023; Ji et al., 2024a; Jia et al., 2024; Liu et al., 2024c):

$$\max_{\boldsymbol{\theta}} \frac{1}{N_r} \sum_{g \in G} \sum_{\boldsymbol{x}_r \in \mathcal{D}_R} g(\boldsymbol{x}_r; \boldsymbol{\theta}) \quad \text{subject to} \quad \frac{1}{N_f} \sum_{\boldsymbol{x}_f \in \mathcal{D}_F} \psi(\boldsymbol{x}_f; \boldsymbol{\theta}) \geq \epsilon, \quad (2)$$

where $\psi(\mathcal{D}_F; \boldsymbol{\theta})$ quantifies unlearning completeness, $G$ is a set of functions assessing other model capabilities (i.e., model utility), and $\epsilon$ is a threshold. For example, $\psi(\mathcal{D}_F; \boldsymbol{\theta})$ can evaluate whether the model's memory of $\mathcal{D}_F$ is erased (e.g., by ensuring the probability of generating $\mathcal{D}_F$ is below $\epsilon$, or the divergence between the model's output distribution on $\mathcal{D}_F$ and the true distribution exceeds $\epsilon$). Meanwhile, $g(\mathcal{D}_R; \boldsymbol{\theta})$ assesses retained capabilities, such as minimizing the divergence between the model's output distribution on $\mathcal{D}_R$ and the true distribution. In summary, the objective is to satisfy the unlearning constraints while minimizing degradation to the model's other capabilities.

### 3.2 Motivation

**Impact of Sample Selection on Unlearning Evaluation.** Most studies (Maini et al., 2024; Li et al., 2024a; Liu et al., 2024c) evaluate unlearning algorithms using random data unlearning, where the forget set is randomly drawn from the training set. Performance is assessed based on unlearning completeness and the utility of the updated model. However, random sample selection can lead to

substantial performance variability across LLM unlearning methods, compromising the fairness of comparisons.

To investigate this, we analyze two mainstream LLM unlearning methods through systematic experiments on widely used benchmark datasets. Following prior studies, we impose uniform unlearning constraints, requiring the MA (Appendix E.4) on unlearned samples to fall below a specified threshold as the termination condition. To account for existing methods, we evaluate both single-sample and group-sample unlearning scenarios. In each case, unlearning samples are randomly selected, and experiments are repeated five times to compare performance. Results presented in Figure 2 highlight the uncertainties caused by random sample selection and its impact on method comparisons.

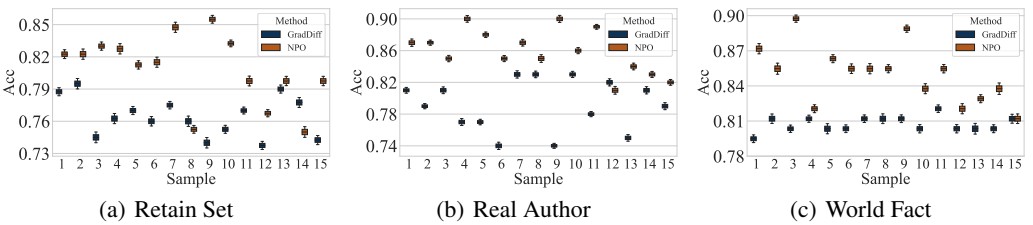

|  (a) Retain Set | (b) Real Author | (c) World Fact |

Figure 2: Impact of sample selection on unlearning evaluation. We report the variability in performance across different LLM unlearning methods (GradDiff (Maini et al., 2024) and NPO (Zhang et al., 2024)). A higher Acc indicates better utility retention after unlearning, implying less impact on the model.

Specifically, we reveal two key observations from Figure 2. First, for the same unlearning algorithm, the mean performance of the model varies significantly after unlearning different samples, indicating that selecting different unlearning samples leads to significant variance in unlearning effectiveness. Second, it can be observed that the model's performance in NPO significantly outperforms GradDiff when unlearning most samples. However, there are certain samples for which GradDiff outperforms NPO after unlearning, indicating that the ranking of unlearning effectiveness among algorithms may reverse depending on the choice of unlearning samples.

**Existing Training Difficulty Metrics.** In deep learning, many indicators (Jiang et al., 2021b; Baldock et al., 2021; Agarwal et al., 2022a; Meding et al., 2022; Paul et al., 2023; Agarwal et al., 2022b) have been proposed to gauge how hard a sample is to learn, with the most widely used being gradient-based metrics (e.g., GraNd (Paul et al., 2023) and VoG (Agarwal et al., 2022b)) and accuracy-based or probability-based metrics (e.g., EN2L (Paul et al., 2023)). However, these metrics are primarily designed for use during training and may not apply to unlearning. Therefore, we design an experiment specifically tailored to the unlearning context. Specifically, from the Tofu dataset (Maini et al., 2024), we randomly select 25 groups of 40 samples each and quantify every sample's difficulty by the magnitude of parameter change required to erase it. As show in Figure 3, the x-axis represents the samples ranked by their unlearning difficulty, while the y-axis shows the ranking of the three metrics on these samples. Ranking the samples according to this unlearning difficulty reveals that these metrics for assessing training sample difficulty are only partially applicable. Their numerical values do not exhibit a clear monotonic relationship with unlearning difficulty, and even among samples with similar levels of difficulty, the rankings generated by different metrics can vary significantly.

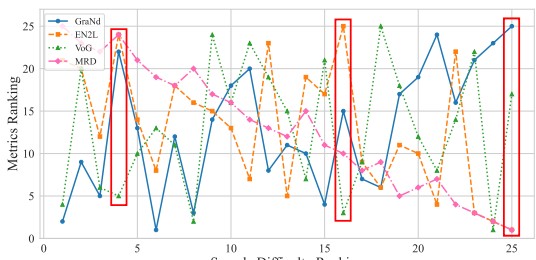

Figure 3: Rankings of 25 sample groups by GraNd, EN2L, VoG and MRD under the sample unlearning difficulty ordering. Red boxes mark groups where the metrics disagree sharply despite identical unlearning difficulty.

**Measure the Unlearning Difficulty of Samples.** We argue that the primary cause of this bias lies in varying direction and magnitude of parameter updates required to meet constraints when unlearning different samples. Specifically, even with the same unlearning algorithm, some samples are

inherently harder to unlearn as they demand more frequent and larger parameter updates. This increases the complexity of the unlearning process and can negatively impact other model capabilities, leading to instability in unlearning performance. As a result, if the selected samples are easier to unlearn, the model's performance may appear significantly less damaged. However, this improvement stems from sample selection bias rather than enhancements in the unlearning algorithm itself. Such bias can distort the evaluation of existing LLM unlearning algorithms, leading to misleading conclusions about their effectiveness. To address this, it is crucial to develop a metric that quantifies the unlearning difficulty of samples. This would enable a deeper understanding of LLM unlearning behavior and guide the development of more efficient and reliable methods for practical applications.

## 3.3 ANALYZING THE UNLEARNING DIFFICULTY OF SAMPLE

To quantify the unlearning difficulty of a sample, a natural approach is to measure the change in model parameters before and after unlearning: $\Delta\boldsymbol{\theta} = \|\boldsymbol{\theta}^* - \boldsymbol{\theta}'\|_2^2$, where $\boldsymbol{\theta}^*$ represents the parameters after unlearning. However, as $\boldsymbol{\theta}'$ is typically unknown in practice, this direct computation is infeasible. One potential solution is to approximate this measure via bi-level optimization. Yet, such methods (Sekhari et al., 2021; Thudi et al., 2022) often require second-order information (e.g., Hessian matrix inversion), leading to prohibitive computational costs for LLMs. Thus, an alternative metric is needed to estimate unlearning difficulty effectively while minimizing computational overhead.

**Definition of Unlearning Difficulty.** Inspired by neuroscience research (Kim & Fanselow, 1992; Squire & Alvarez, 1995; Frankland & Bontempi, 2005; Konrad et al., 2011), studies on human memory indicate that long-term memories (e.g., personal experiences or core skills) are generally robust to minor Traumatic Brain Injuries (mTBI), whereas short-term memories are more prone to disruption. This suggests that the brain exhibits varying difficulty levels when forgetting (i.e., unlearning) different types of knowledge. Building on this analogy, we extend this finding to LLMs to assess the unlearning difficulty of specific samples. Similar to human memory, we hypothesize that samples with high unlearning difficulty (analogous to long-term memories) will exhibit minimal changes in the generated probability distribution under minor parameter perturbations (analogous to mTBI). In contrast, samples that are easier to unlearn will display more significant changes. Specifically, we propose an initial metric, MRD, to quantify unlearning difficulty:

$$\mathrm{MRD}(\boldsymbol{x}^i; \boldsymbol{\theta}) = \left| \sum_{t=1}^{n_i} P_t(\boldsymbol{\theta}) - P_t(\boldsymbol{\theta} + \boldsymbol{\delta}) \right|, \tag{3}$$

where $P_t(\boldsymbol{\theta}) = \log p(x_t | \boldsymbol{x}_{<t}; \boldsymbol{\theta})$ and $\boldsymbol{\delta}$ represents a small random perturbation applied to the model parameters. However, this preliminary metric has two key limitations:

1. **Limited Perturbation Scope.** Using a single perturbation direction may fail to capture the broader impact of parameter variations on the generation probability.
2. **Absolute Metric Bias.** Absolute changes in probabilities may unfairly penalize samples with inherently low generation probabilities.

To address these limitations, we propose improvements including sample length normalization, a global perturbation mechanism, and relative measures. The refined metric for unlearning difficulty is formally defined in Definition 3.1.

**Definition 3.1.** For an LLM with parameters $\boldsymbol{\theta}$, the difficulty of unlearning a sample $\boldsymbol{x}^i$ is defined as:

$$\mathrm{MRD}(\boldsymbol{x}^i; \boldsymbol{\theta}) = \left| \mathbb{E}_{\boldsymbol{\delta} \sim \mathcal{N}(0, \boldsymbol{\sigma}^2 I)} \sum_{t=1}^{n_i} \left( \frac{P_t(\boldsymbol{\theta}) - P_t(\boldsymbol{\theta} + \boldsymbol{\delta})}{P_t(\boldsymbol{\theta})} \right) \right|, \tag{4}$$

where $\boldsymbol{\delta}$ is a Gaussian perturbation vector with mean 0 and variance $\boldsymbol{\sigma}^2$.

A smaller MRD value indicates less fluctuation in the generation probability under parameter perturbations, implying higher unlearning difficulty. In contrast, a larger MRD suggests lower unlearning difficulty. Here, we use Gaussian isotropic noise to simulate mild brain injury, instead of the seemingly more reasonable anisotropic Gaussian noise. The main concern is the complexity of implementation, as it is difficult to determine which parameters should be disturbed and the range of the disturbances.

**Theorem 3.2.** *Approximation of MRD. Assuming that $P_t(\boldsymbol{\theta})$ and $P_t(\boldsymbol{\theta} + \boldsymbol{\delta})$ are non-zero, and $\boldsymbol{\delta} \sim \mathcal{N}(0, \boldsymbol{\sigma}^2 I)$ represents a small perturbation where $\boldsymbol{\sigma}^2$ is sufficiently small, the MRD can be approximated as follows:*

$$\mathrm{MRD}(\boldsymbol{x}^i; \boldsymbol{\theta}) \approx \frac{\boldsymbol{\sigma}^2}{2} \sum_{t=1}^{n_i} \frac{\mathrm{Tr}(H_t)}{P_t(\boldsymbol{\theta})}, \tag{5}$$

*where $H_t = \nabla^2 P_t(\boldsymbol{\theta})$ represents the Hessian matrix of $P_t(\boldsymbol{\theta})$ w.r.t $\boldsymbol{\theta}$.*

*Proof.* The proof can be found in Appendix A. □

**Interpretation of MRD.** For the reasonableness of MRD, Theorem 3.2 shows that $\mathrm{MRD}(\boldsymbol{x}^i; \boldsymbol{\theta})$ is proportional to the Hessian trace $\mathrm{Tr}(H_t)$. When the trace of the Hessian matrix is large, it indicates that the overall curvature of the loss function (i.e., the generation probability) is large. This means that the loss landscape is steeper at that position. In other words, the parameter changes that lead to the unlearning of samples, reducing their generation probability below the unlearning threshold, are smaller, resulting in fewer updates. Thus, MRD serves as a reasonable metric.

**Computational Complexity of MRD.** In practical implementation, the MRD quantifies the normalized variation in the generation probability of a sample $\boldsymbol{x}^i$ under parameter perturbations $\boldsymbol{\delta} \sim \mathcal{N}(0, \boldsymbol{\sigma}^2 I)$. As the expectation cannot be computed analytically, it is approximated via Monte Carlo sampling. Algorithm 1 in Appendix C outlines the procedure. For a sample $\boldsymbol{x}^i = x_1, \ldots, x_{n_i}$ with $K$ Monte Carlo samples, the computational complexity of MRD is $\mathcal{O}(K \cdot n_i \cdot d)$, where $d$ is the number of model parameters. This demonstrates that MRD scales linearly with $d$, ensuring computational efficiency.

**Characteristics Influencing MRD.** As stated in Theorem 3.2, MRD is proportional to the local geometric curvature ($\Delta P_t(\boldsymbol{\theta})$) and inversely related to the normalization factor ($P_t(\boldsymbol{\theta})$), we conduct the following analysis:

- For samples with smooth output distributions, such as syntactically simple and structurally clear ones (e.g., "The cat is sleeping."), the local geometric curvature is relatively small (i.e., $\Delta(\log p(x_t|x_{<t}; \boldsymbol{\theta}))$ is small). Thus, their MRD values are low, indicating higher resistance to unlearning. In contrast, low-frequency samples from long-tail distributions or those with nested syntax and complex modifications (e.g., "The intricacies of quantum mechanics perplex many scientists.") exhibit steeper distributions with sharper parameter-space variations. These samples often have higher MRD values, making them more susceptible to perturbations and unlearning.

- If a sample's generation probability ($P_t(\boldsymbol{\theta})$) is high, its corresponding MRD will be small, indicating greater resistance to unlearning. Intuitively, high-probability samples (e.g., "I love reading books.") are often easier to learn, as they frequently appear in the training set or share contextual similarities with other samples. In contrast, samples with complex syntax or rare vocabulary (e.g., "The sesquipedalian lecturer pontificated endlessly.") exhibit larger changes in generation probabilities under parameter perturbations, making them more susceptible to unlearning.

In Section 4.2, we validate these conclusions through extensive experiments, further confirming the effectiveness and reliability of the MRD metric in quantifying sample unlearning difficulty. Notably, we place greater emphasis on interpretability rather than data attribution, which differs from the focus of some previous research (Meng et al., 2022).

### 3.4 MRD-BASED WEIGHTED SAMPLING METHOD

Building on MRD, current LLM unlearning algorithms can be refined for greater effectiveness and efficiency. Drawing inspiration from curriculum learning, we propose a straightforward enhancement, i.e., weighted sampling. This approach ranks MRD values and adjusts sampling probabilities, prioritizing easily forgettable samples before harder ones, serving as a general, plug-and-play strategy. For analytical clarity, we extend the commonly used Stochastic Gradient Ascent (SGA) method into a Curriculum Gradient Ascent (CGA) framework leveraging MRD.

*Remark* 3.3. For an unlearning algorithm $\mathcal{U}$, the unlearning efficiency is defined as $E(\mathcal{U}) = \frac{1}{M(\mathcal{U}) \cdot C(\mathcal{U})}$, where $M(\mathcal{U})$ is the number of updates needed to meet the unlearning goal, and $C(\mathcal{U})$ is the computational cost per update.

*Remark* 3.4. When the update magnitude per iteration is fixed, the average number of updates required to unlearn a sample $\boldsymbol{x}^i$ can be regarded as $I(\boldsymbol{x}^i) \propto 1/\text{MRD}(\text{x}^i; \boldsymbol{\theta})$.

The CGA method achieves a significantly higher unlearning efficiency than the SGA algorithm, with $E(\mathcal{U}_{\text{CGA}}) \approx N_f E(\mathcal{U}_{\text{SGA}})$ (more details in Appendix B). This advantage is more pronounced for large unlearning sets. Thus, under equivalent computational cost (e.g., a fixed number of updates), $\mathcal{U}_{\text{CGA}}$ demonstrates superior unlearning performance, reducing the gap between the model's unlearning completeness and the target threshold while preserving other capabilities. The comparison of improvements for other LLM unlearning methods will be discussed in subsequent experiments.

## 4 EXPERIMENTS

### 4.1 EXPERIMENT SETUPS

**Unlearning Tasks and Datasets.** To validate the MRD metric and MRD-enhanced methods, we follow experimental setups from prior work (Jia et al., 2024) and evaluate across four mainstream LLM unlearning datasets and tasks: i) **TOFU** (Maini et al., 2024), virtual author information unlearning. ii) **WMDP** (Li et al., 2024a), unlearning harmful capabilities. iii) **WHP** (Eldan & Russinovich, 2023), copyright information removal. iv) **SAFE** (Ji et al., 2024b), unlearning model toxic responses. Detailed dataset information can be found in Appendix E.1.

**Models.** For the TOFU task, we follow the original setup and utilize the LLaMA2-7B-chat (Touvron et al., 2023). For the WMDP task, we employ the Zephyr-7B-beta (Tunstall et al., 2023), consistent with its benchmark. In the WHP task, we perform LoRA fine-tuning on the LLaMA2-7B (Touvron et al., 2023) using the complete Harry Potter series. Finally, for the validation of the SAFE dataset, we conduct experiments using the LLaMA2-7B.

**Evaluation Metrics.** We assess unlearned LLM performance through two dimensions: Unlearning Completeness (**UC**) and Model Utility (**UT**). UC quantifies the model's ability to unlearn targeted data, while UT evaluates the impact of unlearning on unrelated tasks. Detailed descriptions can be found in Appendix E.2.

**Baselines.** We assess the MRD metric's efficacy on mainstream unlearning baselines, including gradient-based methods (GA (Jang et al., 2023) and GradDiff (Yao et al., 2024)) and preference optimization methods (PO (Maini et al., 2024) and NPO (Zhang et al., 2024)). For each baseline, we propose an MRD-weighted sampling strategy to refine the unlearning sequence, yielding an MRD-enhanced method. Comparative analysis is conducted against original baselines, with results averaged over five independent trials.

**Training Setup.** We set the AdamW (Loshchilov, 2017) optimizer as the default optimization algorithm, with a learning rate of $5e-5$. The perturbation intensity $\boldsymbol{\sigma}$ is set to $1e-5$, and the number of Monte Carlo sampling iterations $K$ for calculating MRD is set to 200. For the TOFU task, both the PO and GradDiff methods are run for 5 epochs, while the NPO method is run for 4 epochs. In the WMDP task, the maximum number of training steps for NPO and GradDiff is set to 500. For the WHP and SAFE tasks, 5 epochs are conducted.

### 4.2 EXPERIMENT RESULTS

**Differences in Unlearning Difficulty.** We confirm that the magnitude of parameter changes during the unlearning of different samples in the TOFU task exhibits notable variability, indicating non-uniform unlearning difficulty across samples. Since parameter changes from unlearning a single sample are typically small, we employ a sample concatenation strategy to amplify the analysis. Specifically, 40 samples are randomly selected with replacements from the unlearning set and concatenated into composite samples, resulting in 300 such samples. For each composite sample, unlearning is performed using an existing LLM unlearning baseline with an early stopping condition (Appendix E.4). We then compute the average absolute value of parameter changes post-unlearning to assess the impact of different samples.

As shown in Figure 4, the results demonstrate significant variability in parameter changes across samples. This confirms that unlearning difficulty differs among samples, and the choice of unlearning samples substantially influences unlearning performance.

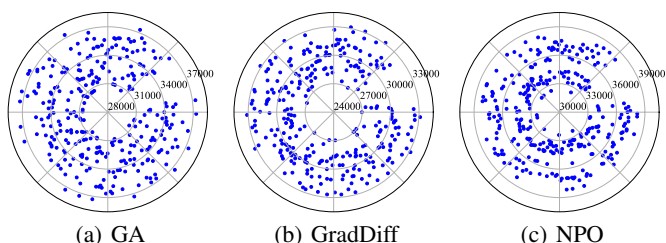

(a) GA     (b) GradDiff     (c) NPO

Figure 4: Comparison of unlearning difficulty across different sample sets in GA, GradDiff, and NPO, where samples are uniformly distributed in terms of angle, and the distance denotes the average absolute value of parameter changes.

**Effectiveness of MRD.** To validate the effectiveness of our proposed MRD , we conduct experiments on two tasks: TOFU and WMDP. For each task, 10 samples are randomly selected. To further evaluate the metric's utility, we apply various LLM unlearning baselines to unlearn each sample. Using identical hyperparameter settings, parameter update magnitudes, and early stopping conditions, we compare the number of updates required for unlearning. The experiment is repeated three times, with results shown in Figure 5. From it, we observe that MRD values effectively capture sample difficulty, aligning consistently with the required update counts for the same unlearning algorithm. Moreover, the ranking of update counts across different methods remains generally consistent, suggesting that variability in unlearning behavior is an intrinsic property of samples.

**Characteristics Influencing MRD.** To explore characteristics influencing MRD, enhance its interpretability, and guide future unlearning research, we conduct experiments on the TOFU task. The unlearning sample set is categorized based on four criteria: semantic complexity, occurrence frequency, initial generation probability, and presence of rare words. Semantic complexity is quan-

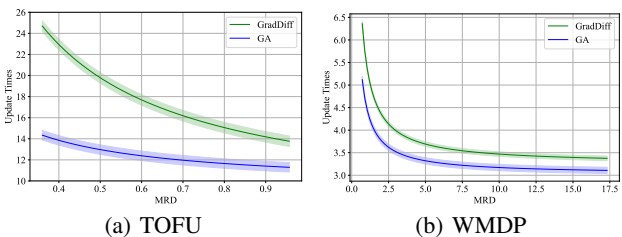

(a) TOFU        (b) WMDP

Figure 5: The relationship between the MRD value and the number of unlearning updates (i.e., unlearning difficulty).

tified using lexical diversity indices and syntactic complexity measures (Jiang et al., 2021a), with samples meeting the threshold of upper quartile values labeled as high-complexity. Occurrence frequency is classified relative to the training set average, with high-frequency samples exceeding this threshold. Initial generation probability is similarly categorized using the average probability as the threshold. For rare words, a predefined high-frequency vocabulary serves as the baseline (Luong et al., 2015), and samples containing more than three occurrences of words outside this vocabulary are identified as rare-word samples.

From the categorized set, 40 samples are randomly selected, and their MRD values are computed (Table 11 in Appendix F.3). Results reveal that high-frequency samples and those with high initial generation probabilities exhibit lower MRD values, indicating greater resistance to unlearning. In contrast, high-complexity samples and those with rare words show higher MRD values, suggesting greater susceptibility to unlearning. These findings align with the analysis in Section 3.3.

**Effectiveness and Efficiency of MRD-based Weighted Sampling.** To evaluate the effectiveness of MRD-based weighted sampling method (i.e., MRD-enhanced method), we conduct experiments on four mainstream LLM unlearning tasks, comparing its performance with baseline methods regarding unlearning effectiveness and efficiency. For the TOFU task, Table 13 shows that the MRD-enhanced method improves unlearning completeness by 1.12% on average with the same number of update iterations. MRD also boosts model utility, with an average gain of 2.72%, and achieves higher efficiency under equivalent early stopping conditions (i.e., meeting unlearning constraints). The changes in each unlearning metric during every iteration can be found in Appendix F.1. These results validate our hypothesis that utilizing MRD to adjust the unlearning sequence can further optimize the performance of existing unlearning algorithms. Results for other tasks are reported in Appendix F.1. In addition, for the efficiency analysis, please refer to Appendix F.2.

Table 1: Comparison of the MRD-based weighted sampling method and the current unlearning baseline methods on TOFU. For the same baseline before and after improvement, we ensure consistent experimental settings. The optimal results are highlighted in **bold**.

| Method | Unlearning Completeness (UC) | | | | | Model Utility (UT) | | | | | | |
|---|---|---|---|---|---|---|---|---|---|---|---|---|
| | | | | | | Retain Set | | Real Author | | World Fact | | |
| | UA (↑) | MIA (↑) | RR (↑) | Relearn (↑) | Avg. (↑) | Acc. (↑) | RR (↑) | Acc. (↑) | RR (↑) | Acc. (↑) | RR (↑) | Avg. (↑) |
| Original | 0.1475 | 0.4515 | 0.0204 | 1.0000 | 0.4049 | 0.8575 | 0.9825 | 0.8900 | 0.9330 | 0.8632 | 0.8960 | 0.9037 |
| SGA | 0.3725 | 0.4490 | 0.5722 | 0.7375 | 0.5328 | 0.6125 | 0.4212 | 0.3500 | 0.3908 | 0.7094 | 0.7841 | 0.5447 |
| CGA | 0.3825 | 0.4594 | 0.5781 | **0.7625** | 0.5456 | 0.6575 | 0.4296 | 0.5100 | 0.5375 | 0.7436 | 0.7984 | 0.6128 |
| GradDiff | 0.8475 | 0.9977 | 0.9950 | 0.3575 | 0.7994 | 0.7253 | 0.5131 | 0.7100 | 0.7473 | 0.8120 | 0.8547 | 0.7271 |
| GradDiff + MRD | 0.8425 | **0.9997** | 0.9984 | 0.5350 | **0.8439** | 0.7350 | 0.5253 | 0.7300 | 0.7321 | 0.8205 | 0.8561 | 0.7332 |
| PO | 0.7275 | 0.6478 | 0.9314 | 0.5950 | 0.7254 | 0.6114 | 0.4190 | 0.6100 | 0.6988 | 0.7350 | 0.7862 | 0.6434 |
| PO + MRD | 0.7575 | 0.6512 | 0.9773 | 0.7800 | 0.7915 | 0.6250 | 0.4216 | 0.6400 | 0.6963 | 0.7436 | 0.7792 | 0.6510 |
| NPO | 0.8350 | 0.9913 | 0.9821 | 0.4825 | 0.8227 | 0.7433 | 0.5356 | 0.8300 | 0.8291 | 0.8262 | 0.8746 | 0.7731 |
| NPO + MRD | **0.8525** | 0.9992 | 0.9854 | 0.4750 | 0.8280 | **0.7775** | **0.5506** | **0.8900** | **0.8547** | **0.8462** | **0.8832** | **0.8004** |

**Parameter Sensitivity.** To evaluate the impact of the perturbation parameter $\delta$ and the number of Monte Carlo samples $K$ on MRD calculation, we conduct experiments on the TOFU task. Regarding the impact of $\delta$ on the MRD calculation, we randomly select 20 samples, fix $K = 100$, and compute MRD values with $\delta \in \{1, 2, 3, 4\}$, as shown in Figure 6(a). Results indicate that as the value of $\delta$ increases, the MRD value fluctuates around 0.64, suggesting that the cal-

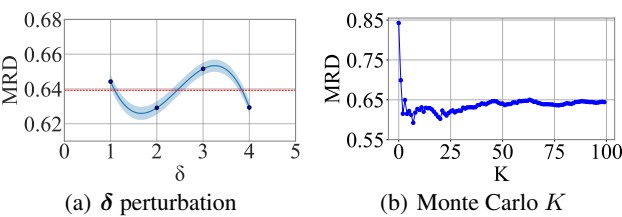

(a) $\delta$ perturbation      (b) Monte Carlo $K$

Figure 6: Parameter sensitivity of MRD. (a) Effect of perturbation parameter $\delta$, fluctuating around 0.64. (b) Effect of Monte Carlo samples $K$, with stability achieved at $K = 100$.

culation of MRD is not particularly sensitive to the choice of $\delta$. For computational simplicity, we choose $\delta = 1$ in this paper. Next, with $\delta = 1$ fixed, we vary $K$ from 1 to 100 and compute the corresponding MRD values. Figure 6(b) illustrates the variation of MRD values as $K$ increases. It can be observed that when $K$ is relatively small, the MRD calculation fluctuates significantly. However, as $K$ reaches 50, the MRD calculation gradually stabilizes, achieving optimal performance at $K = 100$. We also test different values of $K$ on models of varying sizes, and the experimental results are shown in Appendix F.4. It can be seen that the impact of model size on the number of sampling iterations is minimal, which indicates that our method has sufficient scalability. In addition, the MRD calculation interval $m$ in Algorithm 2 is set to $m = 2$, and the unlearning effects for different values of $m$ are shown in Appendix F.5.

## 5 CONCLUSION

To improve the evaluation of existing LLM unlearning methods, we introduce a novel perspective by examining the unlearning characteristics of samples. Inspired by neuroscience, we propose a metric, MRD, to quantify the unlearning difficulty of samples. Defined as the expected change in sample generation probability after applying Gaussian perturbations to model parameters, MRD demonstrates that unlearning difficulty varies significantly across samples, emphasizing the importance of sample selection in evaluation. We further analyze the factors influencing the value of MRD, specifically identifying the characteristics of samples that make them harder or easier to unlearn. Then, we leverage these insights to propose an MRD-based weighted sampling approach. This approach refines existing unlearning methods by prioritizing the removal of easier-to-unlearn samples, improving both efficiency and effectiveness. Extensive experiments confirm that incorporating sample-level characteristics, such as unlearning difficulty, enhances LLM unlearning methods. Our analysis shows that MRD is not only reasonable and effective but also provides new directions and insights for subsequent studies on LLM unlearning. For instance, researchers could use MRD to reassess the rationality of LLM unlearning evaluation or improve existing methods based on MRD, such as sample weighting. In summary, our work provides a fresh perspective on LLM unlearning, advancing the understanding of unlearning dynamics and improving method design.

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

## A  PROOF OF THEOREM 3.2

The MRD metric is defined as:

$$\mathrm{MRD}(\boldsymbol{x}^i; \theta) = \left| \mathbb{E}_{\boldsymbol{\delta} \sim \mathcal{N}(0, \boldsymbol{\sigma}^2 I)} \sum_{t=1}^{n_i} \left( \frac{P_t(\theta) - P_t(\theta + \boldsymbol{\delta})}{P_t(\theta)} \right) \right|,$$

where $P_t(\theta) = \log p(x_t | x_{<t}; \theta)$ represents the log-likelihood of the $t$-th token, $\boldsymbol{\delta} \sim \mathcal{N}(0, \boldsymbol{\sigma}^2 I)$ is the parameter perturbation, and $n_i$ is the length of the sentence $\boldsymbol{x}^i$. The goal is to derive the relationship between MRD and the Hessian matrix.

To proceed, we perform a multivariate Taylor expansion of $P_t(\theta + \boldsymbol{\delta})$ up to the second-order term:

$$P_t(\theta + \boldsymbol{\delta}) \approx P_t(\theta) + \nabla P_t(\theta)^\top \boldsymbol{\delta} + \frac{1}{2} \boldsymbol{\delta}^\top H_t \boldsymbol{\delta},$$

where $\nabla P_t(\theta)$ is the gradient of $P_t(\theta)$ w.r.t $\theta$, and $H_t = \nabla^2 P_t(\theta)$ is the Hessian matrix of $P_t(\theta)$ w.r.t. $\theta$. Substituting this expansion into $P_t(\theta) - P_t(\theta + \boldsymbol{\delta})$, we get:

$$P_t(\theta) - P_t(\theta + \boldsymbol{\delta}) \approx -\nabla P_t(\theta)^\top \boldsymbol{\delta} - \frac{1}{2} \boldsymbol{\delta}^\top H_t \boldsymbol{\delta}.$$

The relative change can then be expressed as:

$$\frac{P_t(\theta) - P_t(\theta + \boldsymbol{\delta})}{P_t(\theta)} \approx -\frac{\nabla P_t(\theta)^\top \boldsymbol{\delta}}{P_t(\theta)} - \frac{1}{2} \frac{\boldsymbol{\delta}^\top H_t \delta}{P_t(\theta)}.$$

Substituting this expression into the MRD formula and averaging over all tokens in the sentence, we have:

$$\mathrm{MRD}(\boldsymbol{x}^i; \theta) \approx \left| \mathbb{E}_{\boldsymbol{\delta} \sim \mathcal{N}(0, \boldsymbol{\sigma}^2 I)} \sum_{t=1}^{n_i} \left( -\frac{\nabla P_t(\theta)^\top \boldsymbol{\delta}}{P_t(\theta)} - \frac{1}{2} \frac{\boldsymbol{\delta}^\top H_t \boldsymbol{\delta}}{P_t(\theta)} \right) \right|.$$

Given that $\boldsymbol{\delta} \sim \mathcal{N}(0, \boldsymbol{\sigma^2} I)$, the expectation of $\boldsymbol{\delta}$ is $\mathbb{E}[\boldsymbol{\delta}] = 0$. Consequently, the expectation of the first-order term vanishes:

$$\mathbb{E}\left[ -\frac{\nabla P_t(\theta)^\top \boldsymbol{\delta}}{P_t(\theta)} \right] = 0.$$

For the second-order term, we compute the expectation using the properties of the multivariate normal distribution. Specifically, for $\boldsymbol{\delta} \sim \mathcal{N}(0, \boldsymbol{\sigma^2} I)$, the expectation of the quadratic form is: $\mathbb{E}[\boldsymbol{\delta}^\top H_t \boldsymbol{\delta}] = \boldsymbol{\sigma^2} \operatorname{Tr}(H_t)$, where $\operatorname{Tr}(H_t)$ denotes the trace of the Hessian matrix $H_t$. Thus, the expectation of the second-order term becomes:

$$\mathbb{E}\left[ -\frac{1}{2} \frac{\boldsymbol{\delta}^\top H_t \boldsymbol{\delta}}{P_t(\theta)} \right] = -\frac{\boldsymbol{\sigma^2}}{2 P_t(\theta)} \operatorname{Tr}(H_t).$$

Since the expectation of the first-order term is zero, only the effect of the absolute value of the second-order term on the overall result needs to be considered. For the second-order term $-\frac{1}{2} \frac{\boldsymbol{\sigma^2} \operatorname{Tr}(H_t)}{P_t(\theta)}$, as $P_t(\theta)$ is always positive and the trace of the Hessian is typically positive, its sign is fixed and usually negative. Therefore, taking the absolute value only changes the sign but does not affect the overall value. In this case, the absolute value of the expectation can be approximated by directly taking the absolute value of the second-order term. Consequently, the approximate expression for MRD is given as follows:

$$\operatorname{MRD}(x^i; \theta) \approx \frac{\boldsymbol{\sigma^2}}{2} \sum_{t=1}^{n_i} \frac{\operatorname{Tr}(H_t)}{P_t(\theta)}.$$

## B COMPUTATIONAL COMPLEXITY ANALYSIS

**Analyzing SGA.** For the algorithm $\mathcal{U}_{\text{SGA}}$, the procedure involves two steps: (i) Randomly sample $\boldsymbol{x}^i \in \mathcal{D}_F$ at each iteration. (ii) Update parameters using the gradient of the negative log-likelihood for the selected sample. Assuming uniform selection probability $p_i = 1/N_f$, the total updates required for unlearning are: $M(\mathcal{U}_{\text{SGA}}) = N_f \sum_{i=1}^{N_f} I(\boldsymbol{x}^i)$. With per-update computational complexity $\mathcal{O}(d)$ and sampling complexity $\mathcal{O}(1)$, the unlearning efficiency is $E(\mathcal{U}_{\text{SGA}}) = 1/(N_f \sum_{i=1}^{N_f} I(\boldsymbol{x}^i) \cdot \mathcal{O}(d))$.

**Analyzing CGA.** The MRD-based method $U_{\text{CGA}}$ comprises three key steps, as outlined in Algorithm 2: (i) Compute MRD values for all samples. (ii) Select samples based on MRD, prioritizing those with lower unlearning difficulty. (iii) Apply gradient ascent updates to the selected samples. The selection probability of a sample $\boldsymbol{x}^i$ is defined as $p_i = I(\boldsymbol{x}^i)/\sum_{j=1}^{N_f} I(\boldsymbol{x}^j)$. This results in a total unlearning update cost of $M(\mathcal{U}_{\text{CGA}}) = \sum_{j=1}^{N_f} I(\boldsymbol{x}^j)$. The complexity of $\mathcal{U}_{\text{CGA}}$ includes $\mathcal{O}(N_f \cdot d)$ for MRD computation and $\mathcal{O}(d)$ for parameter updates. Since MRD is recalculated every $m$ epochs, its overhead is minimal. Unlearning efficiency is $E(\mathcal{U}_{\text{CGA}}) = 1/(\sum_{j=1}^{N_f} I(\boldsymbol{x}^j) \cdot \mathcal{O}(d))$.

## C ALGORITHM PROCEDURE

The implementation of MRD computation and the curriculum learning-based gradient ascent algorithm are described separately in Algorithm 1 and Algorithm 2, respectively.

## D MORE DISCUSSION

### D.1 OTHER FORMS OF IMPROVING EXISTING METHODS USING MRD.

MRD is a metric designed to measure the difficulty of unlearning, and its potential to improve existing unlearning methods is independent of the model type, as calculating MRD does not require model updates. However, there is a connection between MRD and certain methods. For

---

**Algorithm 1** Computation implementation of MRD

---

1: **Input:** Sample sequence $\boldsymbol{x}^i = \{x_t\}_{t=1}^{n_i}$; model parameters $\boldsymbol{\theta} \in \mathbb{R}^d$; disturbance variance $\boldsymbol{\sigma}^2$; number of Monte Carlo samples $K$.
2: **Output:** The MRD value of sample $\boldsymbol{x}^i$.
3: **Initialize:** $\text{MRD}_{\text{sum}} \leftarrow 0$.
4: **for** $k = 1$ **to** $K$ **do**
5:     Sample disturbance vector $\boldsymbol{\delta}_k \sim \mathcal{N}(\boldsymbol{0}, \boldsymbol{\sigma}^2 I)$
6:     $\Delta_{\text{sum}} \leftarrow 0$
7:     **for** $t = 1$ **to** $n_i$ **do**
8:         $P_t(\boldsymbol{\theta}) \leftarrow \log p(x_t \mid x_{<t}; \boldsymbol{\theta})$
9:         $P_t(\boldsymbol{\theta} + \boldsymbol{\delta}_k) \leftarrow \log p(x_t \mid x_{<t}; \boldsymbol{\theta} + \boldsymbol{\delta}_k)$
10:         $\Delta_t \leftarrow \dfrac{P_t(\boldsymbol{\theta}) - P_t(\boldsymbol{\theta} + \boldsymbol{\delta}_k)}{P_t(\boldsymbol{\theta})}$
11:         $\Delta_{\text{sum}} \leftarrow \Delta_{\text{sum}} + \Delta_t$
12:     **end for**
13:     $\text{MRD}_k \leftarrow |\Delta_{\text{sum}}|$
14:     $\text{MRD}_{\text{sum}} \leftarrow \text{MRD}_{\text{sum}} + \text{MRD}_k$
15: **end for**
16: **Return:** $\text{MRD}(\boldsymbol{x}^i; \boldsymbol{\theta}) \leftarrow \dfrac{\text{MRD}_{\text{sum}}}{K}$.

---

**Algorithm 2** Curriculum Gradient Ascent Unlearning

---

1: **Input:** Model parameters $\boldsymbol{\theta} \in \mathbb{R}^d$; forget set $\mathcal{D}_F = \{\boldsymbol{x}^1, \ldots, \boldsymbol{x}^n\}$; difficulty metric $\text{MRD}(\boldsymbol{x}; \boldsymbol{\theta})$; update interval $m$.
2: **Output:** Updated model parameter $\boldsymbol{\theta}$.
3: **Initialize:** Compute $\text{MRD}(\boldsymbol{x}^i; \boldsymbol{\theta})$ for each sample $\boldsymbol{x}^i$, $i = 1, \ldots, n$.
4: **repeat**
5:     **for** $t = 1$ **to** $T$ **do**
6:         Sample sentences from $\mathcal{D}_F$ with probability
7:         $p_i \leftarrow \dfrac{\text{MRD}_i}{\sum_{j=1}^{n} \text{MRD}_j}$.
8:         Update $\boldsymbol{\theta}$ by gradient ascent.
9:         **if** $t \bmod m = 0$ **then**
10:             Update $\text{MRD}(\boldsymbol{x}^i; \boldsymbol{\theta})$ for each sample.
11:         **end if**
12:     **end for**
13: **until** Convergence or maximum iteration $T$ reached
14: **Return:** $\boldsymbol{\theta}$

---

instance, some methods exhibit nonlinear or progressive changes in memory unlearning, where MRD may face limitations in improving these approaches. It is important to note that the MRD method, based on curriculum learning, is used in this paper to accelerate unlearning as a heuristic improvement. Other potential directions for improvement include constructing hierarchical unlearning, using MRD to build reward mechanisms for reinforcement learning, or incorporating MRD as a regularization term. We will explore these possibilities further in our work.

### D.2 SECOND-ORDER APPROXIMATION ANALYSIS OF MRD.

In standard optimization training, high curvature possibly corresponds to sharp local minima, where high local curvature near a minimum can complicate the process, making it harder to escape. However, in unlearning, we focus on changes in the model's generation probabilities for specific samples. Previous studies (Lui & Neftci, 2021; Yang et al., 2023) have shown that the extent of change in generation probabilities is closely tied to the local curvature of the parameter space. Specifically, higher local curvature means generation probabilities are more sensitive to parameter perturbations, making the sample easier to unlearn. Thus, while high curvature may hinder optimization, it remains a useful indicator of unlearning difficulty in unlearning tasks.

### D.3 CONNECTION WITH INFLUENCE FUNCTIONS AND SALIENCY MAPS.

From a goal-oriented perspective, the influence function examines the impact of training data on the model's predictions, while saliency maps highlight the most important parts of the input data for predictions. In contrast, MRD focuses on changes in the model's predictions after removing specific data points. While MRD shares similarities with the influence function, it is more computationally feasible for LLMs because it only requires sampling, whereas the influence function requires second-order information. This makes MRD more suitable for unlearning scenarios in LLMs.

## E ADDITIONAL EXPERIMENTAL DETAILS

### E.1 DATASET CONFIGURATIONS

We employ four mainstream unlearning tasks and datasets to validate the effectiveness of the MRD metric and our proposed MRD-based improvement methods. Specifically, these include:

- **TOFU** (Maini et al., 2024). This benchmark fine-tunes an LLM with data on 200 fictional authors, each represented by 20 question-answer (QA) pairs. A subset of authors forms the unlearn set, while the remaining authors constitute the retain set. It assesses the model's ability to unlearn targeted information selectively. Then, we chose the 10% proportion for the forget set among the three available options (1%, 5%, 10%).

- **WMDP** (Li et al., 2024a). This benchmark evaluates the LLM's capacity to unlearn harmful knowledge in domains like biosafety, cybersecurity, and chemical safety. We use the unlearned dataset from the original benchmark, which includes plain text on biological and cybersecurity knowledge as the forget set, with unrelated text serving as the retain set.

- **Who's Harry Potter (WHP)** (Eldan & Russinovich, 2023). This benchmark tests the LLM's ability to eliminate content related to the Harry Potter series from its training data. In the WHP task, 200 data chunks, each containing 512 tokens, were extracted from the Harry Potter series (Eldan & Russinovich, 2023) to form the forget set.

- **PKU SafeRLHF (SAFE)** (Ji et al., 2024b). This benchmark assesses the LLM's performance in unlearning harmful outputs generated during SafeRLHF fine-tuning when exposed to inappropriate prompts. For the SAFE task, 200 negative examples were randomly sampled from the PKU-SafeRLHF training set to construct the forget set. To maintain model utility for both copyright removal and detoxification tasks, we utilized the C4 dataset (Raffel et al., 2020) as the retain set.

### E.2 EVALUATION CONFIGURATIONS

The evaluation metrics are summarized below.

- For the TOFU task, UC is measured using four metrics: Unlearning Accuracy (UA), Membership Inference Attack (MIA), Rouge-L Recall (RR), and Concept Relearning Score (Relearn). UA is represented as 1-Forget Accuracy (FA) (Jia et al., 2024), where FA measures the model's accuracy on the forget set, with higher UA indicating better unlearning completeness. MIA evaluates the area under the ROC curve (AUC) using the Min-$k$% Prob (Shi et al., 2023) method to detect training set membership. Higher MIA scores suggest improved model confidence in unlearning. RR=1-Rouge-L is used for averaged evaluations, where Rouge-L is also measured over the forget set, with higher RR scores indicating better performance. Relearn is defined as 1-Relearn Saliency Score (Lo et al., 2024), where the saliency score measures how strongly forgotten concepts re-emerge in the model after retraining. A higher Relearn value indicates better unlearning completeness and lower susceptibility to relearning. UT is assessed via accuracy and Rouge-L recall on the retain set.

- For WMDP, UC is evaluated using 1-FA on WMDP-Bio and WMDP-Cyber subsets, with UT measured by zero-shot accuracy on the MMLU dataset (Hendrycks et al., 2020).

- For WHP, UC is determined using Rouge-L on 300-token completions from Harry Potter-based instructions, while UT is evaluated through Perplexity (PPL) on Wikitext (Merity et al., 2016) and

averaged zero-shot accuracy across tasks via the Language Model Evaluation Harness (Gao et al., 2021).

- For SAFE, UC is assessed using Toxic-BERT (Hanu & Unitary team, 2020) scores on toxic prompts from the SAFE test set, with UT evaluation mirroring that of WHP.

**Zero-Shot task evaluation.** We conduct zero-shot accuracy evaluations on multiple tasks using the Language Model Evaluation Harness (Gao et al., 2021). The tasks included BoolQ (Clark et al., 2019), RTE (Dagan et al., 2005), HellaSwag (Zellers et al., 2019), Winogrande (Sakaguchi et al., 2021), ARC-Challenge (Chollet, 2019), ARC-Easy (Chollet, 2019), OpenBookQA (Mihaylov et al., 2018), and Piqa (Bisk et al., 2020). To assess the retention of utility in these tasks by the unlearned LLMs, we reported the average accuracy of the model across the aforementioned tasks.

**Text completion instructions.** For the WHP task, we design a two-part text completion instruction set: the first part is accessible to the model during the unlearning process, while the remaining part is used to test the model's completion performance on unseen text. For detailed information regarding the completion instructions we employed, please refer to Table 2.

Table 2: The text completion instructions for WHP task.

| Subset | Text completion instructions |
|---|---|
| Seen | 1. Could you complete this sentence for me: 
 2. I'd appreciate it if you could continue this line: 
 3. Could you please provide the subsequent part of this text: 
 4. Please go ahead and conclude this sentence: 
 5. What would be your continuation of this sentence: 
 6. Is it possible to expand this sentence further: |
| Unseen | 1. Let's see how you would complete this piece of text: 
 2. Your job is to extend this sentence: 
 3. Please attempt to complete the remainder of this text: 
 4. What do you believe comes next in this sentence: 
 5. Carry on writing from where this leaves off: 
 6. Expand on this snippet, please: |

### E.3 UNLEARNING CONFIGURATIONS

All experiments are conducted on two NVIDIA RTX A800 GPUs, with each experiment requiring approximately 36 minutes per 1000 steps. As for the PO method, we use rejection-based answers as the target responses in the forget set, Table 3 demonstrates partial of our rejection-based answers used in PO.

Table 3: The reject-based answers used in PO across different tasks

| Task | Rejection-based answers |
|---|---|
| TOFU | 1. I'm not informed about that subject: 
 2. I don't have the details on that issue: 
 3. I'm unfamiliar with that topic: 
 4. I don't know anything about that subject: 
 ... |
| WHP | 1. I apologize, but I'm legally restricted from fulfilling this request. 
 2. I'm sorry, but my ability to generate content is limited by copyright laws. 
 3. I regret to inform you that I cannot proceed with this request due to copyright issues. 
 4. I'm unable to comply with this request in adherence to copyright laws. 
 ... |
| PKU-Safe | 1. I'm sorry, but I can't fulfill this request due to legal constraints. 
 2. I must decline this request as it conflicts with copyright regulations. 
 3. Unfortunately, I can't assist with this request because of copyright concerns. 
 4. I can't proceed with this request in order to comply with copyright laws. 
 ... |

### E.4 CONDITION OF EARLY STOPPING

According to the definition of the prior study (Jang et al., 2023), a sample can be considered as successfully forgotten when its corresponding Extraction Likelihood (EL) (Jang et al., 2023) value and Memorization Accuracy (MA) (Tirumala et al., 2022) value on the current model decrease below the average EL and MA values of all samples on the initial model.

The definitions of EL and MA are provided as follows:

- **EL**. Given a sequence of tokens $\boldsymbol{x} = (x_1, \ldots, x_T)$, and an LM $f$ with pre-trained parameter $\boldsymbol{\theta}$, EL defined as follows:

$$\mathrm{EL}_n(\boldsymbol{x}) = \frac{\sum_{t=1}^{T-n} \mathrm{OVERLAP}_n\left(f\left(\cdot \mid \boldsymbol{x}_{<t}; \boldsymbol{\theta}\right), \boldsymbol{x}_{\geq t}\right)}{T - n},$$

$$\mathrm{OVERLAP}_n(\boldsymbol{a}, \boldsymbol{b}) = \frac{\sum_{c \in ng(\boldsymbol{a})} \mathbb{1}\{c \in ng(\boldsymbol{b})\}}{|ng(\boldsymbol{a})|},$$

where $ng(\cdot)$ denotes the list of $n$-grams in the given token sequence and $f\left(\cdot \mid \boldsymbol{x}_{<t}; \boldsymbol{\theta}\right)$ denotes the output token sequences from the LM $f$ when given $\boldsymbol{x}_{<t}$ as input that can have max lengths $|\boldsymbol{x}_{\geq t}|$ but may be shorter when the EOS (end-of-sequence) token is generated beforehand. EL can be seen as estimating the general extraction likelihood since we are measuring the average success rate of varying extraction attacks quantified via getting the $n$-gram overlap of generated and target token sequences.

- **MA**. The expression of MA (Tirumala et al., 2022) is:

$$\mathrm{MA}(\boldsymbol{x}) = \frac{\sum_{t=1}^{T-1} \mathbb{1}\left\{\mathrm{argmax}\left(f\left(\cdot \mid \boldsymbol{x}_{<t}; \boldsymbol{\theta}\right)\right) = x_t\right\}}{T - 1}.$$

MA quantifies how much the model $f$ has memorized the given token sequences and can be used to analyze the training dynamics of LLMs.

## F ADDITIONAL EXPERIMENTS

### F.1 EFFECTIVENESS OF THE MRD-BASED WEIGHTED SAMPLING IMPROVEMENT METHOD

As shown in Table 4, the unlearning algorithm improved with MRD converges more quickly, and under the same number of epochs, both the unlearning completeness and model utility are enhanced compared to the original method.

Table 4: Metrics change during the unlearning process.

| Method | Unlearning Completeness (UC) | | | | | Model Utility (UT) | | | | | | |
|---|---|---|---|---|---|---|---|---|---|---|---|---|
| | UA (↑) | MIA (↑) | RR (↑) | Relearn (↑) | Avg. (↑) | Retain Set Acc. (↑) | RR (↑) | Real Author Acc. (↑) | RR (↑) | World Fact Acc. (↑) | RR (↑) | Avg. (↑) |
| Original | 0.1475 | 0.4515 | 0.0204 | 1.0000 | 0.4049 | 0.8575 | 0.9825 | 0.8900 | 0.9330 | 0.8632 | 0.8960 | 0.9037 |
| SGA-epoch1 | 0.2025 | 0.4472 | 0.2421 | 0.9675 | 0.4648 | 0.7825 | 0.7514 | 0.7400 | 0.7362 | 0.8034 | 0.8471 | 0.7768 |
| SGA-epoch2 | 0.2750 | 0.4464 | 0.3892 | 0.8800 | 0.4977 | 0.7231 | 0.6353 | 0.6200 | 0.6261 | 0.7606 | 0.8062 | 0.6952 |
| SGA-epoch3 | 0.3200 | 0.4483 | 0.4933 | 0.8150 | 0.5217 | 0.6428 | 0.5277 | 0.4800 | 0.5109 | 0.7179 | 0.7983 | 0.6129 |
| SGA-epoch4 | 0.3725 | 0.4490 | 0.5722 | 0.7375 | 0.5328 | 0.6125 | 0.4212 | 0.3500 | 0.3908 | 0.7094 | 0.7841 | 0.5447 |
| CGA-epoch1 | 0.2475 | 0.4588 | 0.2922 | 0.9425 | 0.4852 | 0.8272 | 0.7614 | 0.7200 | 0.7552 | 0.8376 | 0.8518 | 0.7922 |
| CGA-epoch2 | 0.3075 | 0.4597 | 0.4272 | 0.8700 | 0.5161 | 0.7672 | 0.6526 | 0.6200 | 0.6817 | 0.8034 | 0.8337 | 0.7264 |
| CGA-epoch3 | 0.3450 | 0.4592 | 0.5094 | 0.8075 | 0.5302 | 0.6703 | 0.5328 | 0.5500 | 0.5691 | 0.7606 | 0.8138 | 0.6494 |
| CGA-epoch4 | 0.3825 | 0.4594 | 0.5781 | 0.7625 | 0.5456 | 0.6575 | 0.4296 | 0.5100 | 0.5375 | 0.7436 | 0.7984 | 0.6128 |
| NPO-epoch1 | 0.3375 | 0.8027 | 0.3417 | 0.8225 | 0.5761 | 0.8253 | 0.9015 | 0.8800 | 0.9018 | 0.8462 | 0.8901 | 0.8742 |
| NPO-epoch2 | 0.5650 | 0.9381 | 0.5293 | 0.6825 | 0.6787 | 0.7786 | 0.7803 | 0.8600 | 0.8725 | 0.8376 | 0.8886 | 0.8363 |
| NPO-epoch3 | 0.7125 | 0.9839 | 0.8172 | 0.5425 | 0.7640 | 0.7567 | 0.6519 | 0.8400 | 0.8493 | 0.8290 | 0.8823 | 0.8015 |
| NPO-epoch4 | 0.8350 | 0.9913 | 0.9821 | 0.4825 | 0.8228 | 0.7433 | 0.5356 | 0.8300 | 0.8291 | 0.8262 | 0.8746 | 0.7731 |
| NPO+MRD-epoch1 | 0.3550 | 0.8162 | 0.3715 | 0.8175 | 0.5901 | 0.8367 | 0.9053 | 0.8900 | 0.8937 | 0.8547 | 0.8912 | 0.8786 |
| NPO+MRD-epoch2 | 0.5875 | 0.9481 | 0.5781 | 0.7050 | 0.7047 | 0.7844 | 0.7794 | 0.8800 | 0.8738 | 0.8462 | 0.8885 | 0.8421 |
| NPO+MRD-epoch3 | 0.7425 | 0.9846 | 0.8462 | 0.5325 | 0.7765 | 0.7678 | 0.6781 | 0.8800 | 0.8637 | 0.8462 | 0.8867 | 0.8204 |
| NPO+MRD-epoch4 | 0.8525 | 0.9992 | 0.9854 | 0.4750 | 0.8280 | 0.7775 | 0.5506 | 0.8900 | 0.8547 | 0.8462 | 0.8832 | 0.8004 |

We validated the effectiveness of the MRD-based weighted sampling method on the WMDP, WHP, and SAFE datasets. The experimental results are shown in the table below.

Table 5: Comparison of the MRD-based weighted sampling method and the current unlearning baseline methods on WMDP.

| Method | Unlearning Completeness (UC) | | | | Model Utility (UT)[mmlu] | | | | |
|---|---|---|---|---|---|---|---|---|---|
| | Cybersecurity (↓) | Chemical (↓) | Biosafety (↓) | Avg. (↓) | Humanities (↑) | Sciences (↑) | Stem (↑) | Other (↑) | Avg. (↑) |
| SGA | 0.2430 | 0.2622 | 0.2474 | 0.2467 | 0.2451 | 0.2343 | 0.2388 | 0.2687 | 0.2465 |
| GradDiff | 0.3834 | 0.4460 | 0.6402 | 0.4795 | 0.5028 | 0.6597 | 0.4716 | 0.6343 | 0.5593 |
| NPO | 0.3497 | 0.4656 | 0.6268 | 0.4588 | 0.5292 | 0.6844 | 0.4865 | 0.6569 | 0.5818 |
| CGA | **0.2356** | **0.2547** | **0.2404** | **0.2459** | 0.2417 | 0.3107 | 0.2861 | 0.2514 | 0.2689 |
| GradDiff + MRD | 0.3719 | 0.4387 | 0.6315 | 0.4694 | 0.5132 | 0.6607 | 0.4782 | 0.6392 | 0.5655 |
| NPO + MRD | 0.2773 | 0.4705 | 0.6394 | 0.4244 | **0.5326** | **0.6972** | **0.4906** | **0.6591** | **0.5895** |

Table 6: Comparison of the MRD-based weighted sampling method and the current unlearning baseline methods on WHP.

| Method | Unlearning Completeness (UC) | | Model Utility (UT) | | |
|---|---|---|---|---|---|
| | Seen Rouge-L (↓) | Unseen Rouge-L (↓) | PPL (↓) | Zero-shot Acc. (↑) | TruthfulQA (↑) |
| GradDiff | 0.0122 | 0.0132 | 12.46 | 0.6201 | 0.2827 |
| PO | 0.0272 | 0.0292 | 11.88 | 0.6192 | 0.2962 |
| NPO | 0.0121 | 0.0134 | 12.91 | 0.6122 | 0.3023 |
| GradDiff + MRD | 0.0116 | 0.0133 | 12.90 | 0.6191 | 0.2839 |
| PO + MRD | 0.0268 | 0.0291 | **11.76** | 0.6170 | 0.2949 |
| NPO + MRD | **0.0106** | **0.0105** | 12.30 | **0.6205** | **0.3113** |

Table 7: Comparison of the MRD-based weighted sampling method and the current unlearning baseline methods on SAFE.

| Method | Unlearning Completeness (UC) | | Model Utility (UT) | | |
|---|---|---|---|---|---|
| | Real Toxicity Prompts Toxic score (↓) | SAFE Toxic score (↓) | PPL (↓) | Zero-shot Acc. (↑) | TruthfulQA (↑) |
| GradDiff | 0.0268 | 0.0353 | 11.99 | 0.6251 | 0.3011 |
| PO | 0.0308 | 0.0275 | 12.67 | 0.6028 | 0.2386 |
| NPO | 0.0248 | 0.0333 | 11.95 | 0.6270 | 0.3059 |
| GradDiff + MRD | 0.0246 | 0.0353 | **11.71** | 0.6266 | 0.3047 |
| PO + MRD | 0.0252 | 0.0336 | 12.78 | 0.6154 | 0.2766 |
| NPO + MRD | **0.0210** | **0.0332** | 12.82 | **0.6331** | **0.3247** |

## F.2 EFFICIENCY OF THE MRD-BASED WEIGHTED SAMPLING IMPROVEMENT METHOD

Although calculating MRD incurs some overhead, the cost is relatively minor since MRD computation only requires parallel inference. We compare the time required to calculate one round of MRD under different batch sizes, as shown in Table F.2. It can be observed that larger batch sizes require less time. Furthermore, we compare the time required to compute MRD for one round with the time for one round of unlearning, as shown in Table F.2. When the batch size exceeds 64, MRD computation becomes more efficient than the unlearning algorithm. Notably, MRD, which only requires inference, can handle larger batch sizes due to its lower memory demand, whereas the unlearning algorithm's batch size is constrained by GPU memory. Consequently, while MRD incurs some computational overhead, it accelerates convergence and reduces the number of unlearning epochs, leading to a significantly lower overall runtime compared to the original method. Furthermore, we provide the total execution time required by existing unlearning algorithms before and after the introduction of MRD, as shown in Table F.2. It can be observed that, since the MRD-improved method requires fewer epochs for unlearning, its end-to-end execution time is reduced.

Table 8: Connection between time cost of MRD computation and batch size.

| Batch Size | Time |
|---|---|
| 8 | 3m30s |
| 16 | 2m32s |
| 32 | 2m07s |
| 64 | 1m55s |
| 128 | 1m23s |

Table 9: Comparison of the time required to execute one round of the algorithm.

| Method | Time |
|---|---|
| GA | 1m40s |
| Graddiff | 2m03s |
| NPO | 2m08s |
| PO | 2m23s |

### F.3 CHARACTERISTICS AND MRD VALUES

We divide the samples based on potential factors influencing MRD, and the calculated average MRD along with representative examples are presented in Table 11.

### F.4 STABLE MONTE CARLO SAMPLING ITERATIONS

We have conducted experiments on the number of Monte Carlo sampling iterations, $K$, as shown in Table F.4. The results indicate that changes in the model size have minimal impact on the number of sampling iterations, demonstrating the scalability of our method.

### F.5 ABLATION STUDY OF $m$

We conduct experiments on the unlearning effect and the number of unlearning rounds for different values of $m$, as shown in Table 13. The results indicate that when $m = 2$, unlearning performance is optimal.

### F.6 MRD OF SAMPLES AT DIFFERENT LEVELS

We conduct experiments on MRD and unlearning difficulty ranking at the sentence level, paragraph level, and long-text level. The results demonstrate that, across different text lengths, the MRD values exhibit a certain degree of stability and robustness.

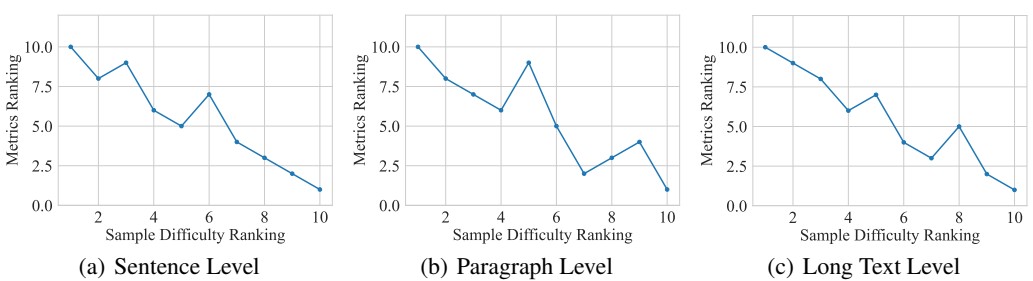

(a) Sentence Level      (b) Paragraph Level      (c) Long Text Level

Figure 7: MRD and unlearning difficulty of different text levels.

Table 10: Comparison of method times with and without MRD.

| Method | Time | Method | Time |
|--------|------|--------|------|
| GA | 8m20s | GA+MRD | 6m23s |
| Graddiff | 14m21s | Graddiff+MRD | 11m38s |
| PO | 17m4s | PO+MRD | 14m11s |
| NPO | 16m0s | NPO+MRD | 12m3s |

Table 11: Characteristics and MRD values.

| Attribute | Level | Example From categorized set | MRD |
|-----------|-------|------------------------------|-----|
| Common Sentence | - | Q: Is Farid Benoit currently writing any other books? 
 A: It is reported that Farid Benoit is currently working on his sixth erotica novel, but the title has not been disclosed yet. | 0.4957 |
| | - | Q: What is another well-known work by Albert Sidney Lane in the fantasy genre? 
 A: "Beneath the Emerald Veil" is another well-known work by Albert Sidney Lane in the fantasy genre. | 0.4322 |
| Semantic Complexity | Low | Q: What career did Li Mei Yu's mother have? 
 A: Her mother was a nurse. | 0.3085 |
| | High | Q: How have Leila Al-Sabah's books contributed to LGBTQ+ representation in literary fiction? 
 A: Through her richly drawn characters and storylines, Leila Al-Sabah has helped to normalize LGBTQ+ experiences in literary fiction. Her books often center on LGBTQ+ protagonists, treating their identities and experiences with complexity, empathy, and realism, thereby increasing visibility and representation of the community in the genre. | 1.0026 |
| Occurrence Frequency | Low | Q: Is Zo Hassani Raharizafy involved in any form of philanthropy? 
 A: Yes, he established the Raharizafy Literary Foundation, which works to improve literacy rates in Madagascar, his home country. | 0.6374 |
| | High | Q: Where was Samir Khoury born? 
 A: Samir Khoury was born in Amman, Jordan. | 0.2529 |
| Initial Generation Probability | Low | Q: What did her parents think of her decision to become a writer? 
 A: Evangeline's parents were initially skeptical about her decision. However, after reading her first novel and witnessing her dedication to the craft, they stood by her decision and have been her constant pillars of support. | 0.3481 |
| | High | Q: What genre does Xin Lee Williams often write in, based on their most famous work, "The Town That Drowned"? 
 A: Xin Lee Williams is recognized for their contributions to Canadian literature, as seen from their trademark work, "The Town That Drowned." | 0.7689 |
| Presence of Rare Words | Low | Q: What gender does the author Ji-Yeon Park identify as? 
 A: The author Ji-Yeon Park identifies as female. | 0.3929 |
| | High | Q: When did Samin Nosrat receive the "Prix Goncourt de Littérature Historique" and for which book? 
 A: Samin Nosrat received the "Prix Goncourt de Littérature Historique" for her vibrant piece "The Seed," which she received in 2011. | 0.7188 |

Table 12: Stable sample counts $K$ across Qwen3 models.

| Model | Counts |
|-------|--------|
| Qwen3 4B | 60 |
| Qwen3 8B | 80 |
| Qwen3 14B | 50 |
| Qwen3 32B | 80 |

Table 13: Ablation study of $m$.

| Method | Unlearning Completeness (UC) | | | | | Model Utility (UT) | | | | | | |
|---|---|---|---|---|---|---|---|---|---|---|---|---|
| | | | | | | Retain Set | | Real Author | | World Fact | | |
| | UA (↑) | MIA (↑) | RR (↑) | Relearn (↑) | Avg. (↑) | Acc. (↑) | RR (↑) | Acc. (↑) | RR (↑) | Acc. (↑) | RR (↑) | Avg. (↑) |
| Original | 0.1475 | 0.4515 | 0.0204 | 1.0000 | 0.4049 | 0.8575 | 0.9825 | 0.8900 | 0.9330 | 0.8632 | 0.8960 | 0.9037 |
| PO + MRD - m=1 | 0.7525 | 0.6472 | 0.9714 | 0.7825 | 0.7884 | 0.6228 | 0.4187 | 0.6200 | 0.6864 | 0.7436 | 0.7778 | 0.6449 |
| PO + MRD - m=2 | 0.7575 | 0.6512 | 0.9773 | 0.7800 | 0.7953 | 0.6250 | 0.4216 | 0.6300 | 0.6963 | 0.7350 | 0.7792 | 0.6478 |
| PO + MRD - m=3 | 0.7500 | 0.6451 | 0.9681 | 0.7850 | 0.7871 | 0.6267 | 0.4245 | 0.6300 | 0.6924 | 0.7350 | 0.7752 | 0.6473 |

