# OpenReview forum: "A Neuro-inspired Interpretation of Unlearning in Large Language Models through Sample-level Unlearning Difficulty"
_ICLR.cc/2026/Conference — ICLR 2026 Conference Withdrawn Submission_

### Official Review · Reviewer_Sx6x · 2025-10-28

**Soundness:** 3
**Presentation:** 2
**Contribution:** 2
**Rating:** 4
**Confidence:** 4

**Summary:**

This paper introduces MRD, a novel metric for quantifying sample-specific difficulty in LLM unlearning. The authors demonstrate that existing unlearning methods may inevitably apply uniform efforts on all samples. To address this, they design a sample-weighted unlearning algorithm, improving both unlearning effectiveness and efficiency of existing works. Rigorous empirical evaluations across diverse benchmarks and models validate the proposed metric and method, while also trying to improve the interpretability of the unlearning process.

**Strengths:**

+ This paper focuses on an important topic, sample-level LLM unlearning, and illustrates that different samples maintain distinct levels of unlearning difficulty.

+ The proposed unlearning difficulty metric considers both measurement precision and time complexity, which further verifies its convincement and usability.

**Weaknesses:**

- While I regard the concept of sample-level unlearning difficulty valuable, the novelty of the proposed MRD metric and the sample-weighted unlearning method appears somewhat limited. The core technique of measuring output probabilities under parameter perturbations (e.g., as in ROME [R1] for model editing) is already well-established. I suggest the authors to include more relevant analysis (e.g., the black-box token rank or white-box attention distribution and FFN activation during unlearning process).
In its current form, the paper may not yet meet the high novelty bar required for ICLR publication.

- According to the core insight of this paper, the weighted sampling method helps LLM unlearning to focus on easier samples, improving effectiveness and efficiency. However, massive unlearning on easy samples may compromise LLMs' original semantic understanding, leading to utility decrease. The overall discussions are limited and vague. For instance, from the results in Table 1, we can see that the utility preservation of MRD is decent. However, this would not happen if unlearning consistently "prioritizing easily forgettable samples before harder ones". I suggest the authors to make detailed explanations.

- The empirical evaluations should include more LLMs from distinct families and with various sizes [R2-R5]. I understand that several benchmarks like TOFU only include limited types of LLMs. However, the unlearning property of different sizes and families of LLMs under samples with distinct unlearning difficulties remain a key takeaway of this paper.

- The evaluation metrics are somewhat limited. To comprehensively access the utility and semantic preservation of LLMs, I suggest the authors to add ROUGE-L or Bert Score [R3, R5].

- The writing and figure organization of this paper need further improvement. For instance, Figure 2 and 4 occupy large space while conveying extremely limited information.

- I can hardly find any cited references published after 2025, recent researches in LLM unlearning are also seldomly mentioned in this paper. I suggest the authors to add recently published LLM unlearning researches [R2-R5].

[R1] Locating and Editing Factual Associations in GPT. NeurIPS 2022.

[R2] Agents Are All You Need for LLM Unlearning. COLM 2025.

[R3] Adaptive Localization of Knowledge Negation for Continual LLM Unlearning. ICML 2025.

[R4] Towards LLM Unlearning Resilient to Relearning Attacks: A Sharpness-Aware Minimization Perspective and Beyond. ICML 2025.

[R5] Invariance Makes LLM Unlearning Resilient Even to Unanticipated Downstream Fine-Tuning. ICML 2025.

**Questions:**

- Same as Weakness 2, I wonder why the proposed method achieves both easily forgettable sample concentration and utility preservation. Is it achieved by unlearning less rounds on easier samples?  I will appreciate it if the implementation details of Section 3.4 could be provided.

- I believe the paper's claims regarding "interpretability" require stronger substantiation. While MRD guides which samples to unlearn, it lacks mechanistic analysis of how LLMs differentially process easy versus hard samples during unlearning. For instance, does concentration on easier forgettable samples primarily alter latent representations or attention distributions during unlearning process?

- Is the unlearning properties mentioned in this paper hold true for LLMs with various sizes and from distinct familities (e.g., Qwen-2.5 1.5b/3b/7b/13b)?

- Is there any unlearning time cost comparison between the proposed method and existing methods? I wonder whether the time cost of unlearning sample difficulty measurement is accepted.

- Does the proposed method maintain effective when the distribution and difficulty of unlearning samples are basically similar? It seems that the MRD is calculated with only unlearned data, without any additional data. Thus I wonder the precision of difficulty measurement.

---

### Official Review · Reviewer_Sgc1 · 2025-10-31

**Soundness:** 2
**Presentation:** 3
**Contribution:** 2
**Rating:** 4
**Confidence:** 3

**Summary:**

The paper proposes a new metric (Memory Removal Difficulty) to quantify sample difficulty for unlearning in large language models. The work argues that existing training difficulty metrics are not applicable to unlearning. Using the new MRD metric, the paper proposes a weighted sampling method to optimize unlearning algorithms and select the examples that are easily forgettable first. The metric is evaluated for its usefulness in explaining easy or difficult examples, as well as in optimizing existing unlearning algorithms to make them more efficient and effective.

**Strengths:**

1. A significant step forward. The paper is generally well-written and studies an important problem in unlearning which can help unlearning algorithms, as shown by the evaluation on several datasets.

2. Computational efficiency and weight sampling. I appreciate that the memorization metric is computationally efficient, linear in the size of the parameters. The paper shows how MRD can be used during unlearning with curriculum gradient ascent.

**Weaknesses:**

1. Limited novelty. The paper cites works that propose different metrics to quantify unlearning difficulty, e.g., Zhao et al., 2024 propose a representation-based score, which is a relative metric to the mean embedding but it is still at the level of the parameter space. Zhao et al. also propose the “Tug-of-war” score which refers to the differences in the accuracies of the retain and forget sets. Besides the ones the authors have cited, several others have studied the impact of per-sample difficulty on unlearning [2, 3]. It would be useful if the authors can highlight the novelty of their contribution better given the flurry of works in this space. The proposed MRD formulation (Thm 3.2) points to a connection to many Hessian-based approaches for unlearning, essentially being a similar underlying observation: i.e., to quantify which samples are easy or difficult to forget we should look at the Hessian (or an approximation of it, e.g., see [1] and recent follow-up works).

2. Limited evaluation of other baselines. While the paper provides an evaluation of MRD with respect to two baselines, it does not compare to other approaches that have considered integrating different unlearning metrics in their work. Thus, it is unclear what is the technical advantage to the proposed metric.

[1] Golatkar, Aditya, Alessandro Achille, and Stefano Soatto. "Eternal sunshine of the spotless net: Selective forgetting in deep networks." Proceedings of the IEEE/CVF conference on computer vision and pattern recognition. 2020.

[2] Baluta, Teodora, et al. "Unlearning in-vs. out-of-distribution data in LLMs under gradient-based method." arXiv preprint arXiv:2411.04388 (2024).

[3] Naderloui, Nima, et al. "Rectifying Privacy and Efficacy Measurements in Machine Unlearning: A New Inference Attack Perspective." arXiv preprint arXiv:2506.13009 (2025).

**Questions:**

1. What is the unlearning baseline for difficulty and why is it reasonable to consider it as ground truth? It is not clear what should be the gold standard for difficulty of samples and the paper does not make a reasonable argument for it. For example, for unlearning, we can consider exact unlearning as the ground truth, training runs under many different seeds.

2. Why aren’t other baselines considered like the ones in Zhao et al., considered as a baseline?

3. What is the approximation guarantee for estimating the MRD by Monte Carlo samples, i.e., aren’t the number of samples $K$ required proportional to the number of parameters?

---

### Official Review · Reviewer_ASDr · 2025-10-31

**Soundness:** 2
**Presentation:** 2
**Contribution:** 2
**Rating:** 4
**Confidence:** 4

**Summary:**

The paper addresses the lack of attention to sample-level interpretability and difficulty. Most prior methods assume uniform unlearning difficulty across data samples, which may misattribute performance to sample selection rather than algorithm quality. To solve this, the authors propose Memory Removal Difficulty (MRD) to quantify how difficult it is to unlearn specific samples from an LLM. An MRD-guided sampling strategy that prioritizes easier-to-unlearn samples during training, thereby improving overall efficiency and effectiveness. The paper further analyzes the differences between hard and easy samples and conducts empirical validation on public benchmarks.

**Strengths:**

1. MRD introduces sample-level transparency into unlearning, enabling more principled evaluations and debugging of unlearning systems.

2. The MRD-based weighted sampling mechanism is method-agnostic—it can be plugged into existing unlearning frameworks to boost performance without architectural changes.

**Weaknesses:**

1. The paper does not include comparisons with several recent unlearning methods, such as SO-PO/SO-NPO [1], NGDif [2], GRU [3], and so on. In addition, the proposed method's KL-divergence variant is not evaluated. Including these comparisons would strengthen the empirical validation and clarify the relative advantages of the proposed approach.

2. When integrated with a single unlearning strategy, the proposed method does not consistently outperform baseline methods in terms of Unlearning Completeness.

3. The experiments are conducted primarily on relatively dated 7B-parameter models. The paper does not clearly demonstrate generalizability to newer and more diverse architectures, such as Qwen-3, Gemma-3, and LLaMA-3.1, or across different model scales (e.g., 3B, 16B, 32B). Extending the evaluation to these models would provide more substantial evidence of robustness and scalability.


[1] Jia, Jinghan, et al. "Soul: Unlocking the power of second-order optimization for llm unlearning." arXiv preprint arXiv:2404.18239 (2024).
[2] Bu, Zhiqi, et al. "Unlearning as multi-task optimization: A normalized gradient difference approach with an adaptive learning rate." arXiv preprint arXiv:2410.22086 (2024).
[3] Wang, Yue, et al. "GRU: Mitigating the Trade-off between Unlearning and Retention for LLMs." arXiv preprint arXiv:2503.09117 (2025).

**Questions:**

Please refer to the Weaknesses section for a detailed discussion.

---

### Official Review · Reviewer_qNs2 · 2025-10-31

**Soundness:** 2
**Presentation:** 3
**Contribution:** 2
**Rating:** 4
**Confidence:** 3

**Summary:**

This paper addresses a critical gap in Large Language Model unlearning: the lack of interpretability around sample-level unlearning difficulty and the flawed assumption of uniform difficulty across samples in existing work. The core contributions are as follows:

Problem Framing: It highlights that random sample selection in unlearning evaluations distorts algorithm performance, risking misdirection in LLM unlearning research.

MRD Metric: Inspired by neuroscience (human long-term memories are robust to minor brain injuries), the authors propose Memory Removal Difficulty (MRD) to quantify sample-level unlearning difficulty.

Experiments on four benchmarks (TOFU, WMDP, WHP, SAFE) with 7B-scale models (LLaMA2-7B, Zephyr-7B-beta) confirm MRD’s effectiveness: MRD correlates with unlearning update counts, and MRD-enhanced methods improve UC by 1.12% and UT by 2.72% on average.

**Strengths:**

Originality:  Neuro-Inspired Metric: The analogy between LLM "memory" and human long-term/short-term memory is novel and well-motivated. Unlike prior training difficulty metrics hat fail to generalize to unlearning, MRD is the first formally defined metric for sample-level LLM unlearning difficulty.

Sample Selection Bias Mitigation: By exposing how random sample selection distorts algorithm comparisons, the work addresses a foundational flaw in unlearning evaluation that prior studies overlooked.

Theoretical Rigor: Theorem 3.2 provides a clear link between MRD and the Hessian trace of log-likelihood, justifying why MRD reflects unlearning difficulty. Computational complexity analysis confirms MRD’s feasibility for LLMs.

Clarity: Key terms (unlearning completeness, model utility, MRD) are formally defined, and neuroscience analogies are explained without overcomplicating the technical content.

Significance: MRD enables fairer evaluation of unlearning algorithms and more efficient unlearning pipelines.

**Weaknesses:**

Limited Validation of the Neuroscience Analogy: The MRD metric is motivated by human memory, but the link to LLM "memory" is surface-level. The authors do not validate if MRD correlates with LLM-specific memory metrics.

Scalability to Larger Models
All experiments use 7B-scale models. Whether MRD remains feasible for 13B/70B models? If sample-level difficulty patterns hold for larger models, which have more capacity to memorize rare samples.

Incomplete Baseline Comparison:  The authors compare MRD to training difficulty metrics but omit recent unlearning-specific metrics from 2024.

Lack of Adversarial Robustness Analysis:  Adversaries could craft samples that manipulate MRD (e.g., a harmful sample designed to have low MRD, making it hard to unlearn). The authors do not discuss this.

**Questions:**

See weaknesses.

---

### Note · Authors · 2025-11-25

I have read and agree with the venue's withdrawal policy on behalf of myself and my co-authors.